# PLA Biocomposites: Evaluation of Resistance to Mold

**DOI:** 10.3390/polym14010157

**Published:** 2021-12-31

**Authors:** Piotr Borysiuk, Krzysztof Krajewski, Alicja Auriga, Radosław Auriga, Izabela Betlej, Katarzyna Rybak, Małgorzata Nowacka, Piotr Boruszewski

**Affiliations:** 1Institute of Wood Sciences and Furniture, Warsaw University of Life Sciences—SGGW, ul. Nowoursynowska 159, 02-776 Warsaw, Poland; krzysztof_krajewski@sggw.edu.pl (K.K.); radoslaw_auriga@sggw.edu.pl (R.A.); izabela_betlej@sggw.edu.pl (I.B.); piotr_boruszewski@sggw.edu.pl (P.B.); 2Faculty of Biotechnology and Animal Husbandry, West Pomeranian University of Technology Szczecin, Janickiego 33, 71-270 Szczecin, Poland; alicja.auriga@tlen.pl; 3Department of Food Engineering and Process Management, Institute of Food Sciences, Warsaw University of Life Sciences—SGGW, 159C Nowoursynowska St., 02-776 Warsaw, Poland; katarzyna_rybak@sggw.edu.pl (K.R.); malgorzata_nowacka@sggw.edu.pl (M.N.)

**Keywords:** PLA, HDPE, biocomposites, mold, bark

## Abstract

Due to the content of lignocellulosic particles, wood plastic composites (WPC) composites can be attacked by both domestic and mold fungi. Household fungi reduce the mechanical properties of composites, while mold fungi reduce the aesthetics of products by changing their color and surface decomposition of the wood substance. As part of this study, the impact of lignocellulosic fillers in the form of sawdust and bark in poly (lactic acid) (PLA)-based biocomposites on their susceptibility to mold growth was determined. The evaluation of the samples fouled with mold fungi was performed by computer analysis of the image. For comparison, tests were carried out on analogous high-density polyethylene (HDPE) composites. Three levels of composites’ filling were used with two degrees of comminution of lignocellulosic fillers and the addition of bonding aids to selected variants. The composites were produced in two stages employing extrusion and flat pressing. The research revealed that PLA composites were characterized by a higher fouling rate by Aspergillus niger Tiegh fungi compared to HDPE composites. In the case of HDPE composites. The type of filler (bark, sawdust) affected this process much more in the case of HDPE composites than for PLA composites. In addition, the use of filler with smaller particles enhanced the fouling process.

## 1. Introduction

The rapidly developing industry of wood-plastic composites (WPC) focuses on the introduction of new material solutions for matrix and fillers. In both cases, the biodegradability of applied raw materials is crucial. Nowadays most WPC composites are produced from polyethylene PE, polypropene PP, or polyvinyl chloride PVC [1]. However, as an alternative can also be used biodegradable poly (lactic acid) PLA or polyhydroxyalkanoate PHA. PHA when exposed to anaerobic conditions slowly decomposes under the influence of bacteria present in the soil, sewage, or silt into water and carbon dioxide. For this reason, it can be applied to manufacture packaging and components with short durability. In turn, PLA does not biodegrade under ordinary conditions of use, so it can be applied in production of components with a long mean life. Furthermore, PLA can be easily disposed of by composting with no harm to the natural environment [2]. Given its features, PLA is used in medicine and industry, replacing conventional petrochemical polymers [3]. However, due to its downsides, such as sensitivity to moisture, susceptibility to aging, limited impact strength, and high rigidity [4], PLA is modified in many ways.

One of the directions of PLA modification is introduction lignocellulosic fillers, such as: wood fibers [5,6,7], wood flour [8,9,10], cork [8], bamboo fiber [11], abaca fibers [12], rubber wood sawdust [13], and bark [14]. The authors generally indicate an improvement in the mechanical properties of PLA composites filled with wood fibers with a filler content of up to 20% [5,7] or 30% [6]. Andrzejewski et al. [8] reported the beneficial effect of cork filler (up to 30%) on the dimensional stability of PLA composites exposed to moisture. The use of wood flour or bark as a filler, on the other hand, deteriorates the resistance of PLA composites to moisture [10,14].

Due to the content of lignocellulosic particles and changing conditions of use WPC composites can be biodegradable [15,16,17,18,19,20,21,22,23]. They are susceptible to domestic and mold fungi attack. Domestic fungi cause changes in the structure and chemical composition of the lignocellulosic particles in the composites. The degradation effect depends on the weight fraction of lignocellulosic particles, their size and type of wood, as well as the possible application of other additives [24]. The degradation of wood particles is reflected in the decrease in the strength of WPC composites [20,23,25,26], and to a large extent this decrease is also caused by changes caused by the moistening and drying of these particles [27]. Mold fungi, in turn, reduce the aesthetics of the products by changing their color and the surface distribution of the wood substance [28]. Fungi have a detrimental effect on the health of humans and animals living in the vicinity of objects attacked by them [29,30]. Schirp et al. [31], Kartal et al. [32], Feng et al. [33] reported that WPC with a higher content and larger sizes of wood particles are more susceptible to mold fungi. The susceptibility also depends on the type of wood used as the filler [34].

This study determines the impact of lignocellulosic fillers (sawdust and bark) in PLA biocomposites on their susceptibility to mold growth. Schirp et al. [31] reported that the influence of coloring and mold fungi on WPC was measured only by the visual method of assessment of microorganisms fouling the material. As part of the research, the mold growth on the samples was assessment using computer analysis of the image [14]. The tests were carried out on analogous WPC composites made of high-density polyethylene (HDPE). Three levels of filling were used with two degrees of granularity of lignocellulosic fillers and the addition of joining additives to selected variants. The composites were produced in two stages process consisting of extrusion and flat pressing.

## 2. Materials and Methods

In this study 36 variants of WPC composite panels were produced based on two types of polymer matrices: polylactic acid—PLA (Ingeo^TM^ Biopolymer 2003D, NatureWorks LLC, Minnetonka, MN, USA) and high-density polyethylene—HDPE (Hostalen GD 7255, Basell Orlen Polyolefins Sp. Z.o.o., Płock, Poland) (Table 1 and Table 2). Two types of lignocellulosic material were used as a filler: coniferous sawdust and conifer bark. Additives used in selected variants were: calcium oxide CaO (Avantor Performance Materials Poland S. A., Gliwice, Poland) in the case of PLA composites, and polyethylene-graft-maleic anhydride MAHPE (SCONA TSPE 2102 GAHD, BYK-Chemie GmbH, Wesel, Germany) in HDPE composites.

The lignocellulosic material obtained from the sawmill was dried to a humidity of 5% and then mechanically ground and sorted into two size variants:

(1)Particles passing through a 2 mm sieve (approx. 10 mesh) and remaining on a 0.49 mm sieve (approx. 35 mesh);(2)Particles passing 0.49 mm sieve (greater than 35 mesh).

The composites were produced in a two stage process:

(1)First, WPC granules with an appropriate formulation were produced (Table 1 and Table 2) using an extruder (Leistritz Extrusionstechnik GmbH, Nürnberg, Germany) (temperatures in individual sections of the extruder were 170–180 °C), the obtained continuous composite band was then ground in a hammer mill.(2)Secondly, the obtained granulate was used to produce plates with nominal dimensions 300 × 300 × 2.5 mm^3^. The process consisted of flat pressing in a mold, using a one-shelf press (AB AK Eriksson, Mariannelund, Sweden) at a temperature of 200 °C and a maximum unit pressing pressure pmax = 1.25 MPa (the pressure during pressing, along with the plasticization of the material, was gradually increased from 0 to pmax). The pressing time was 6 min. After hot pressing, the plates were cooled in the mold for 6 min in the cold press.

The manufactured plates were conditioned for 7 days at 20 ± 2 °C and 65 ± 5% humidity.

### 2.1. Resistance to Moulds

The resistance of materials to molds was performed using the test specimens of dimensions 50 × 50 × 2.5 mm^3^. Test samples were superficially sterilized by spraying all surfaces with 70% alcohol and then placed separately in sterile glass vessels for 24 h at a temperature of 65 °C. After cooling the samples for the next 24 h, test specimens were exposed to pure cultures of Aspergillus niger Tiegh (ATCC:16888) fungus, growing on a 2% MEA nutrient medium (OXOID Ltd., Basingstoke, UK).

The specimens were placed directly into Petri dishes (diameter of 100 mm) on a nutrient agar medium to ensure their good moisture saturation. Inoculation with the fungus was carried out by placing four inoculums, each, approximately 10 mm from every edge of the specimen. Growth of fungus was conducted in incubators chamber—model Thermolyne Type 42000 (ThermoFisher Scientific, Waltham, MA, USA) for 22 days at temperature of 26 °C. Periodically the mold growth on samples was determined by taking high resolution pictures in laboratory photo making cabinet station for documentation purposes. In accordance with the author’s own concept of assessing the degree of contamination (p) of the tested materials by *A. niger*, two parameters of attack by the fungus were determined for each sample. The first parameter of infestation (p_1_) was the percentage of mycelium coverage of the sample surface, calculated in relation to the total area of the sample. The second parameter of infestation (p_2_) determined the percentage of cover of the sample surface by conidial sporangia of the fungus, calculated in relation to the total area of the sample. The final result of the specimen contamination by the fungus (p), was assumed as the value of the sum of the parameters p_1_ and p_2_, calculated according to the equation p = 0.7p_1_ + 0.3p_2_. The numerical factors 0.7 and 0.3, respectively for the parameters p_1_ and p_2_, were adopted on the basis of our own observations and considered as suitable for the parametric determination of the degree of contamination of materials by *A. niger*. The results of the contamination of samples by the fungus was the arithmetic mean value of the results obtained for four replications of each material variant.

The percentage overgrowth of samples was determined with an accuracy of up to 5% with the support of image analysis software ImageJ2 (Fiji v1.52i) [35,36].

### 2.2. Porosity

The porosity and the pore volume in the samples was determined by using the X-ray micro-CT measurements SkyScan 1272 system (Bruker microCT, Kontich, Belgium). To capture high quality imaging pixel resolution of 25.0 μm, 40 keV source voltage, 193 μA current was used. A stack of approximately 1500 flat projection images (1008 × 1008 pixels) was obtained after a 180° rotation with 0.4° steps, which averaged 4 frames for each step.

## 3. Results

At the initial stage (first 3 days) mold growth on PLA composites was significantly influenced by the size of the filler particles (PS) and the interaction between the share of the filler and its particles size (FCxPS), the share of the filler and its type (FCXF), and the proportion of the filler, the particles size of the filler and the type of filler (FCxPSxF). In each of the cases, the influence of factors was significant (the influence percentage of factors ranged from 16.27% to 27.55%). In turn, further growth of mold was to a lesser extent determined by the size of the filler particles, although this effect was still significant (Table 3). In the final stage of fouling the PLA samples, the greatest influence percentage had the share of the filler (FC) and the type of filler (F) as well as the interaction between these factors (FCxF).

In the case of HDPE composites (Table 4), the filler particles size (PS) had the greatest effect on the mold growth over the first 7 days. On the other hand, the further growth was also determined by the share of the filler (FC) and the type of filler (F) as well as the interaction between these factors (FCxF).

The effect of addition the additive and its interaction with the filler was significant in the first five days of exposure to fungi in case of PLA composites (Table 5). For HDPE composites this effect was generally significant throughout the impact of mold fungi (Table 6). Referring to these analyzes, it is worth noting that, except for HDPE composites filled with large sawdust, in the remaining cases, after four days of fouling, the influence of factors not included in this study started to play an important role (error > 20%).

Regardless the type of filler, PLA composites were more susceptible to mold growth than analogous HDPE composites. After 7 days, 90% of PLA composites surface was covered by mold irrespectively of the composition of the specimens (Figure 1). In the case of HDPE composites, 100% surface coverage of the samples was not achieved even after 15 days of exposure (Figure 2). Exemplary images of samples covered by mold fungi are presented in Figure 3, Figure 4, Figure 5 and Figure 6. Composites made on the basis of PLA were characterized by a generally higher porosity of the internal structure compared to analogous materials made on the basis of HDPE (Figure 7).

Application of the additives had a significant effect on the mold growth on both, PLA and HDPE composites (Figure 1a,d, Figure 2a,d, Figure 8 and Figure 9). Irrespectively to the type of filler, the additives increased the rate of mold growth. It is worth adding here that the porosity of the composites in most cases (except for the composite based on polyethylene filled with small bark) generally decreased (Figure 10). 

## 4. Discussion

The higher susceptibility to mold of PLA composites compared to HDPE composites is consistent with existing scientific literature. Zimmermann [37] reports that aliphatic polyesters, including PLA, are more susceptible to microbial degradation than non-hydrolyzable synthetic polymers, such as, inter alia, PE. Maeda et al. [38] showed that fungi of the genus Aspergillus had a hydrolyzing effect on PLA. Porosity, in turn, increases the availability of composite components for microbiological agents and thus increases their susceptibility to degradation [28].

In the case of WPC composites, the addition of additional substances (e.g., a compatibilizer) has a significant impact on their physical and mechanical properties [28]. Yeh et al. [39] revealed that the addition of a compatibilizer has a positive effect on reducing the degradation of WPC composites by fungi throughout improving the reduction of moisture penetration. In the present study, CaO was introduced as an additive—a moisture absorbing and biocidal agent to PLA composites [40], and the MAHPE was applied in HDPE composites [28].

Factors such as type of filler, its contribution and size of particles affected the mold growth regardless the type of matrix PLA or HDPE (Table 3 and Table 4). The interaction between these factors is being significant. It is worth noting that this impact varies depending on the duration of the exposure to the mold. Schirp et al. [31], Kartal et al. [32], Feng et al. [33] reported that WPC with a higher content of larger size wood particles are more susceptible to the influence of mold fungi. Ref. [41] found that an increase in the thermoplastic content in the outer layers from 40% to 50% in particle-polymer boards elongates mold growth (Trichoderma virens) about 3.5 times.

Regardless of the matrix type (PLA or HDPE), bark-filled composites were more susceptible to mold. In relation to the tested materials, it is particularly visible in composites filled with large particles (10–35 mesh). This is probably due to the greater availability of large particles for microorganisms, while smaller particles (less than 10 mesh) are better surrounded by the polymer matrix and thus less accessible to fungi. The influence of the type of filler on WPC susceptibility to mold was also demonstrated by Xu et al. [42], Feng et al. [34] and Feng et al. [43], Valentín et al. [44] stated that pine bark can be an excellent source of nutrients for fungi. So et al. [45], in the study of litter, revealed that bark particles are more susceptible to mold growth than coniferous chips used under the same conditions. It is worth noting that the bark contains more extractives, and Hosseinaei et al. [46] found that reducing their content limits the susceptibility of WPC to mold growth.

Nevertheless of the growth rate results, it should be enhanced that mold fungi reduce the aesthetics of WPC products by changing their color and the surface decomposition of the wood substance [28]. Additionally, objects attacked by fungi have a detrimental effect on the health of humans and animals living in their vicinity [29,30].

## 5. Conclusions

PLA composites are characterized by a higher growth rate by *Aspergillus niger* Tiegh mold fungi compared to HDPE composites.The type of filler (bark, sawdust) had a greater impact on fouling by mold fungi in the case of HDPE composites.Composites filled with bark were characterized by a higher growth rate of mold fungi compared to composites filled with sawdust.In the case of sawdust filler, composites filled with small particles revealed a higher fouling rate.In the case of bark filler, PLA composites displayed a higher fouling rate when filled with large particles, while HDPE composites revealed higher fouling rate when filled with small particles.The introduction of additional substances (CaO in PLA composites and MAHPE in HDPE composites) generally increased the rate of mold growth on the composites.

## Figures and Tables

**Figure 1 polymers-14-00157-f001:**
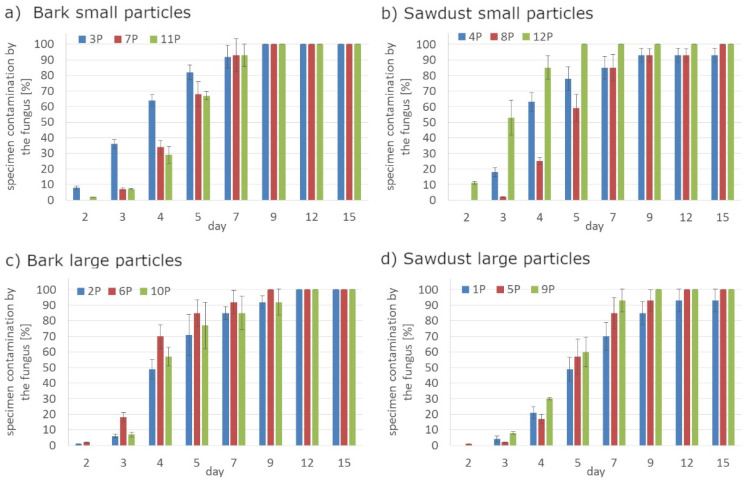
The rate of surface fouling by *Aspergillus niger* Tiegh fungi on PLA composites for: (**a**) bark small particles; (**b**) sawdust particles; (**c**) bark large particles; (**d**) sawdust large particles.

**Figure 2 polymers-14-00157-f002:**
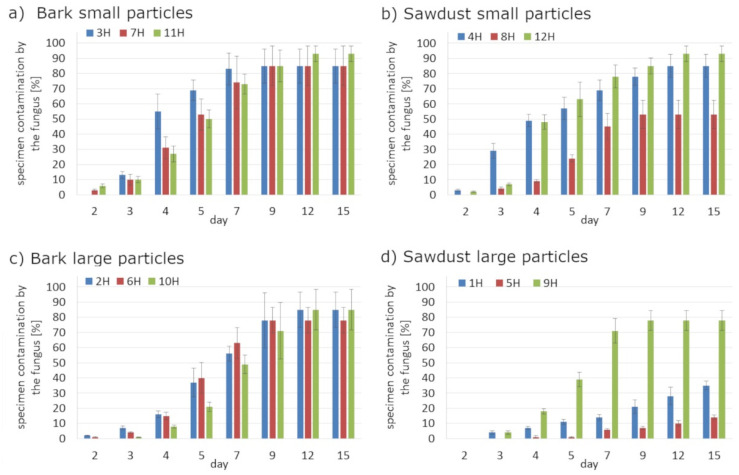
The rate of surface fouling by *Aspergillus niger* Tiegh fungi on HDPE composites for: (**a**) bark small particles; (**b**) sawdust particles; (**c**) bark large particles; (**d**) sawdust large particles.

**Figure 3 polymers-14-00157-f003:**
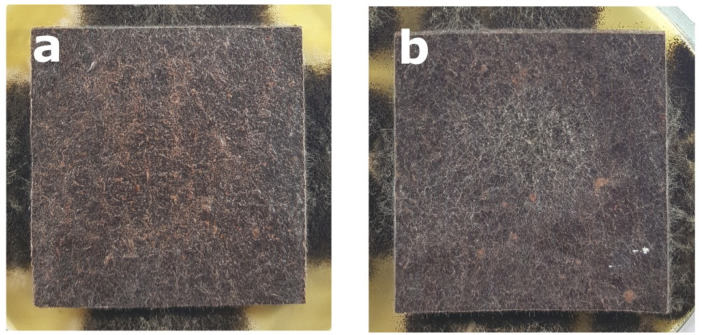
Example of mold growth (*Aspergillus niger* Tiegh) after 7 days of exposure of composite samples with large particle bark filler of 60%, based on a matrix: (**a**) HDPE; (**b**) PLA.

**Figure 4 polymers-14-00157-f004:**
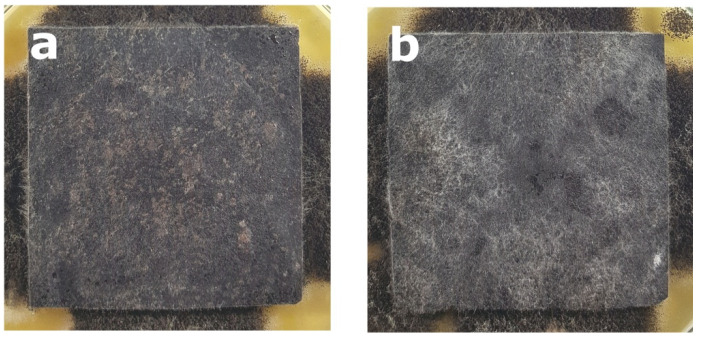
Example of mold growth (*Aspergillus niger* Tiegh) after 7 days of exposure of composite samples with small particle bark filler of 60%, based on a matrix: (**a**) HDPE; (**b**) PLA.

**Figure 5 polymers-14-00157-f005:**
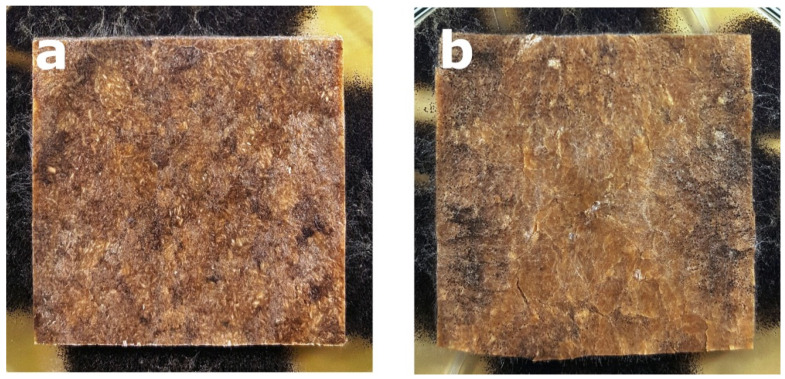
Example of mold growth (*Aspergillus niger* Tiegh) after 7 days of exposure of composite samples with large particle sawdust filler of 60%, based on a matrix: (**a**) HDPE; (**b**) PLA.

**Figure 6 polymers-14-00157-f006:**
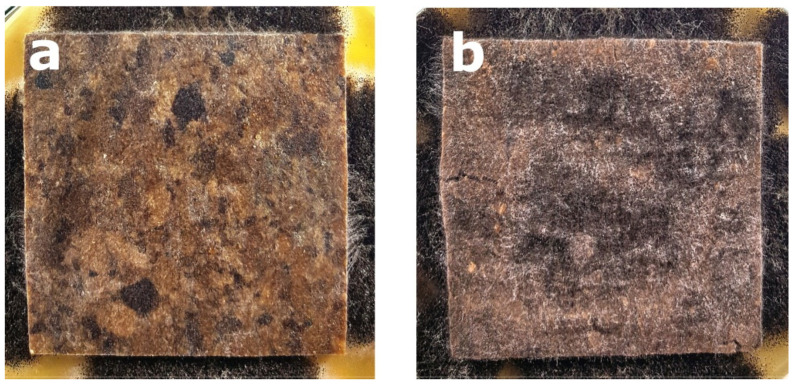
Example of mold growth (*Aspergillus niger* Tiegh) after 7 days of exposure of composite samples with small particle sawdust filler of 60%, based on a matrix: (**a**) HDPE; (**b**) PLA.

**Figure 7 polymers-14-00157-f007:**
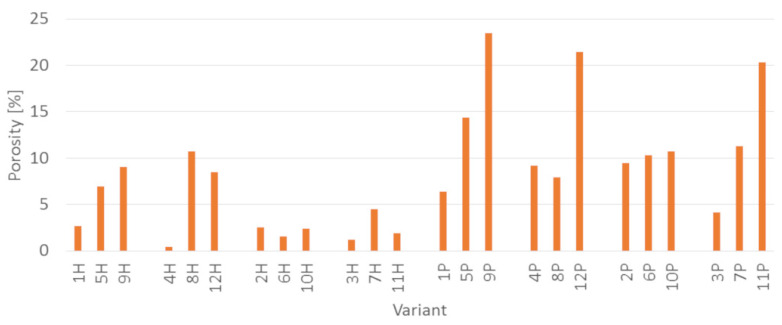
Porosity of the tested PLA and HDPE composites.

**Figure 8 polymers-14-00157-f008:**
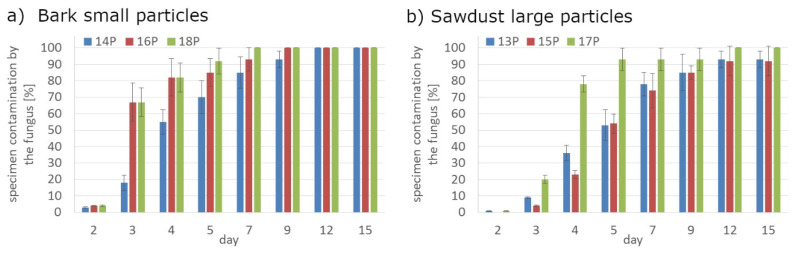
The rate of surface fouling by fungi *Aspergillus niger* Tiegh of PLA composites with CaO addition; (**a**) bark small particles; (**b**) sawdust large particles.

**Figure 9 polymers-14-00157-f009:**
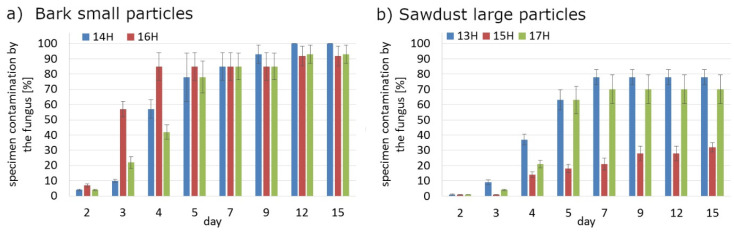
The rate of surface fouling by fungi *Aspergillus niger* Tiegh of HDPE composites with MAHPE addition; (**a**) bark small particles; (**b**) sawdust large particles.

**Figure 10 polymers-14-00157-f010:**
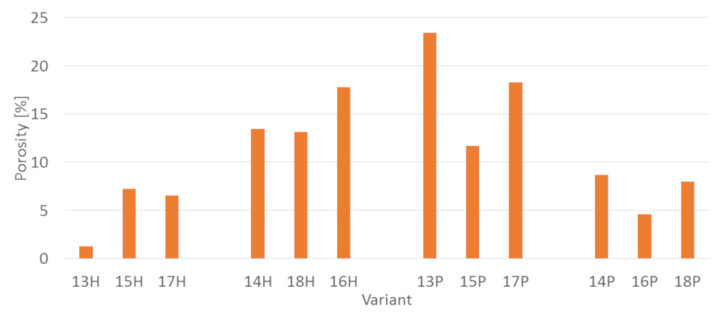
The porosity of the tested PLA and HDPE composites containing additives.

**Table 1 polymers-14-00157-t001:** Composition of individual variants PLA composites.

Variant	Matrix	Share of the Matix (%)	Additvies (CaO) (%)	Share of the Filler [%]	
				Small Particles ˃35 Mesh	Large Particles 10–35 Mesh
1P	PLA	60			40 s
2P	PLA	60			40 b
3P	PLA	60		40 b	
4P	PLA	60		40 s	
5P	PLA	50			50 s
6P	PLA	50			50 b
7P	PLA	50		50 b	
8P	PLA	50		50 s	
9P	PLA	40			60 s
10P	PLA	40			60 b
11P	PLA	40		60 b	
12P	PLA	40		60 s	
13P	PLA	57	3		40 s
14P	PLA	57	3	40 b	
15P	PLA	47	3		50 s
16P	PLA	47	3	50 b	
17P	PLA	37	3		60 s
18P	PLA	37	3	60 b	

s—sawdust, b—bark.

**Table 2 polymers-14-00157-t002:** Composition of individual variants HDPE composites.

Variant	Matrix	Share of the Matix (%)	Additvies (MAHPE) (%)	Share of the Filler [%]	
				Small Particles ˃35 Mesh	Large Particles 10–35 Mesh
1H	HDPE	60			40 s
2H	HDPE	60			40 b
3H	HDPE	60		40 b	
4H	HDPE	60		40 s	
5H	HDPE	50			50 s
6H	HDPE	50			50 b
7H	HDPE	50		50 b	
8H	HDPE	50		50 s	
9H	HDPE	40			60 s
10H	HDPE	40			60 b
11H	HDPE	40		60 b	
12H	HDPE	40		60 s	
13H	HDPE	57	3		40 s
14H	HDPE	57	3	40 b	
15H	HDPE	47	3		50 s
16H	HDPE	47	3	50 b	
17H	HDPE	37	3		60 s
18H	HDPE	37	3	60 b	

s—sawdust, b—bark.

**Table 3 polymers-14-00157-t003:** The influence percentage of individual factors and their interactions affecting the mold growth on PLA composites.

Day	Factors			Interaction between Factors				Error
	FC	PS	F	FCxPS	FCxF	PSxF	FCxPSxF	
2	8.83 ^S^	20.11 ^S^	0.09 ^N^	19.18 ^S^	27.46 ^S^	0.39 ^S^	22.87 ^S^	1.04
3	10.44 ^S^	18.54 ^S^	0.08 ^N^	18.48 ^S^	27.55 ^S^	4.61 ^S^	16.27 ^S^	4.33
4	8.37 ^S^	4.51 ^S^	5.72 ^S^	16.56 ^S^	19.16 ^S^	34.84 ^S^	6.63 ^S^	4.19
5	5.71 ^S^	8.41 ^S^	6.35 ^S^	14.61 ^S^	12.28 ^S^	22.35 ^S^	5.61 ^S^	24.66
7	15.12 ^S^	9.96 ^S^	3.55 ^N^	5.52 ^N^	16.68 ^S^	0.39 ^N^	0.82 ^N^	47.96
9	14.67 ^S^	9.51 ^S^	9.29 ^S^	6.34 ^N^	18.78 ^S^	1.06 ^N^	2.09 ^N^	38.25
12	14.13 ^S^	2.35 ^N^	21.18 ^S^	4.71 ^N^	14.13 ^S^	2.35 ^N^	4.71 ^N^	36.43
15	24.65 ^S^	0.00 ^S^	12.33 ^S^	0.00 ^S^	24.65 ^S^	0.00 ^S^	0.00 ^S^	38.37

FC—filler content; PS—particle size; F—filler; ^N^—no statistical significance (*p* > 0.05); ^S^—statistically significant (*p* < 0.05).

**Table 4 polymers-14-00157-t004:** The influence percentage of individual factors and their interactions affecting the mold growth on HDPE composites.

Day	Factors			Interaction between Factors				Error
	FC	PS	F	FCxPS	FCxF	PSxF	FSxPSxF	
2	2.78 ^S^	24.00 ^S^	10.07 ^S^	13.19 ^S^	11.73 ^S^	0.05 ^N^	31.92 ^S^	6.26
3	24.91 ^S^	35.85 ^S^	0.02 ^N^	9.37 ^S^	9.53 ^S^	1.08 ^S^	13.33 ^S^	5.91
4	16.37 ^S^	51.02 ^S^	1.01 ^S^	10.34 ^S^	14.89 ^S^	0.10 ^N^	1.13 ^S^	5.13
5	9.38 ^S^	43.31 ^S^	8.90 ^S^	4.39 ^S^	24.15 ^S^	0.62 ^N^	0.71 ^N^	8.54
7	11.65 ^S^	29.56 ^S^	15.37 ^S^	4.56 ^S^	23.52 ^S^	1.78 ^S^	4.67 ^S^	8.88
9	13.41 ^S^	18.52 ^S^	24.54 ^S^	2.72 ^S^	17.78 ^S^	6.14 ^S^	5.06 ^S^	11.74
12	21.75 ^S^	16.20 ^S^	26.19 ^S^	1.93 ^S^	12.38 ^S^	9.57 ^S^	3.59 ^S^	8.40
15	23.22 ^S^	15.48 ^S^	25.99 ^S^	1.46 ^S^	12.95 ^S^	8.66 ^N^	3.00 ^S^	9.24

FC—filler content; PS—particle size; F—filler; ^N^—no statistical significance (*p* > 0.05); ^S^—statistically significant (*p* < 0.05).

**Table 5 polymers-14-00157-t005:** Percentage of the compatibilizer influence, filler fraction and their interactions influencing the mold growth on PLA composites.

Day	Bark Large Particles				Bark Small Particles			
	K	FC	KxFC	Error	K	FC	KxFC	Error
2	31.80 ^S^	9.47 ^S^	56.19 ^S^	2.54	0.03 ^N^	33.62 ^S^	62.90 ^S^	3.45
3	26.35 ^S^	58.06 ^S^	11.96 ^S^	3.64	41.85 ^S^	3.39 ^S^	50.32 ^S^	4.45
4	29.96 ^S^	49.03 ^S^	18.69 ^S^	2.31	50.31 ^S^	0.53 ^N^	40.44 ^S^	8.71
5	11.96 ^S^	44.63 ^S^	22.90 ^S^	20.52	20.91 ^S^	2.34 ^N^	45.62 ^S^	31.13
7	0.26 ^N^	46.93 ^S^	10.98 ^N^	41.83	0.00 ^N^	14.96 ^N^	14.86 ^N^	70.19
9	9.28 ^N^	32.64 ^S^	4.73 ^N^	53.34	14.04 ^S^	28.08 ^S^	28.08 ^S^	29.80
12	4.54 ^N^	26.77 ^S^	8.99 ^N^	59.70				
15	4.54 ^N^	26.77 ^S^	8.99 ^N^	59.70				

K—additives; FC—filler content; ^N^—no statistical significance (*p* > 0.05); ^S^—statistically significant (*p* < 0.05).

**Table 6 polymers-14-00157-t006:** Percentage of the compatibilizer influence, filler fraction and their interactions influencing the mold growth on HDPE composites.

Day	Sawdust large Particles				Bark Small Particles			
	K	FC	KxFC	Error	K	FC	KxFC	Error
2	72.87 ^S^	11.21 ^S^	11.21 ^S^	4.70	19.79 ^S^	44.51 ^S^	26.57 ^S^	9.14
3	15.73 ^S^	60.40 ^S^	16.16 ^S^	7.71	28.03 ^S^	31.70 ^S^	37.52 ^S^	2.75
4	44.68 ^S^	29.95 ^S^	22.92 ^S^	2.45	32.75 ^S^	26.81 ^S^	29.88 ^S^	10.56
5	39.23 ^S^	47.53 ^S^	10.07 ^S^	3.18	51.77 ^S^	6.35 ^N^	10.60 ^N^	31.28
7	18.20 ^S^	58.94 ^S^	20.14 ^S^	2.71	15.44 ^N^	4.65 ^N^	4.64 ^N^	75.27
9	15.77 ^S^	61.44 ^S^	19.86 ^S^	2.93	1.92 ^N^	3.78 ^N^	3.80 ^N^	90.51
12	12.85 ^S^	65.79 ^S^	17.79 ^S^	3.57	18.45 ^S^	4.11 ^N^	12.23 ^N^	65.21
15	11.78 ^S^	68.75 ^S^	16.03 ^S^	3.44	18.45 ^S^	4.11 ^N^	12.23 ^N^	65.21

K—additives; FC—filler content; ^N^—no statistical significance (*p* > 0.05); ^S^—statistically significant (*p* < 0.05).

## Data Availability

The data presented in this study are available on request from the corresponding author.

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
