# Peer review of "PLA Biocomposites: Evaluation of Resistance to Mold"

_polymers, 2021, doi:10.3390/polym14010157_

Round 1
Reviewer 1 Report
- In Abstract: It would be better to define abbreviations at first use. Such as WPC, PLA etc.
- Page 1, line 42-45: Please rephrase the sentences.
- This manuscript is required rigorous English editing. Please consider to improve this paper.
- In the abstract, this is written as “the use of filler with smaller particles accelerated the fouling process” whereas author referred literature on Page 2, line 65-66 as “Feng et al. [33] reported that WPC with a higher content and larger sizes of wood particles are more susceptible to mold fungi”. Please explain.
- Table 1: content of the materials are in % or weight? Please confirm. Also, please include used additive name in the table 1and 2.
- Give some reasoning for use of CaO and MAHPE for PLA and HDPE? How it will affect material’s properties?
- Reason behind enhance mold growth should be given in the manuscript.
- Data related to pristine PLA and HDPE should also be included in this work for comparison.
- Some literature references should be included in the manuscript explaining what content makes bark responsible for higher mold growth.
- It is not explained in this manuscript that why particles size is affecting fouling rate differently for PLA and HDPE?
Author Response
We thank for professional comments on the manuscript. The manuscript has been revised according to the comments and suggestions. Each comment and its corresponding response are outlined below, including a sample of manuscript text where the modifications were applied, as well as a reference to page and line numbers in the revised manuscript.
- In Abstract: It would be better to define abbreviations at first use. Such as WPC, PLA etc.
Thank you for the comment, the abbreviations has been defined.
- Page 1, line 42-45: Please rephrase the sentences.
The type of filler (bark, sawdust) affected this process much more in the case of HDPE composites than for PLA composites. In addition, the use of filler with smaller particles enhanced the fouling process.
- This manuscript is required rigorous English editing. Please consider to improve this paper.
The manuscript has been revised and improved.
- In the abstract, this is written as “the use of filler with smaller particles accelerated the fouling process” whereas author referred literature on Page 2, line 65-66 as “Feng et al. [33] reported that WPC with a higher content and larger sizes of wood particles are more susceptible to mold fungi”. Please explain.
For the same % by mass of the filler, an increase in its fragmentation causes that it is more available on the surface, which at the same time makes it more susceptible to fouling. Larger particles are less often dispersed in the matrix and therefore its availability on the composite surface is relatively smaller.
- Table 1: content of the materials are in % or weight? Please confirm. Also, please include used additive name in the table 1and 2.
Content of the materials are in %, the additives CaO and MAHPE have been added in both tables (1 and 2).
- Give some reasoning for use of CaO and MAHPE for PLA and HDPE? How it will affect material’s properties?
“In the case of WPC composites, the addition of additional substances (e.g. a compatibilizer) has a significant impact on their physical and mechanical properties [28]. Yeh et al. (2021) revealed that the addition of a compatibilizer has a positive effect on reducing the degradation of WPC composites by fungi throughout improving the reduction of moisture penetration. In the present study CaO was introduced as an additive - a moisture absorbing and biocidal agent to PLA composites, [40], and the MAHPE was applied in HDPE composites (Klysov 2007).” 283-289. MAHPE – is a standard bonding aid in PE-based composites - improves mechanical and physical properties.
- Reason behind enhance mold growth should be given in the manuscript.
Perhaps this is due to the greater availability of nutrients and the surface structure (greater roughness enhance mold growth)
- Data related to pristine PLA and HDPE should also be included in this work for comparison.
The work investigated composites made of selected thermoplastics and different fillers - the plastics themselves were not the subject of the investigation.
- Some literature references should be included in the manuscript explaining what content makes bark responsible for higher mold growth.
„The influence of the type of filler on WPC susceptibility to mold was also demonstrated by Xu et al. (2015), Feng et al. (2016) and Feng et al. (2019), Valentín et al. (2010) whose stated that pine bark can be an excellent source of nutrients for fungi. SO et al. (1978), in the study of litter, revealed that bark particles are more susceptible to mold growth than coniferous chips used under the same conditions. It is worth noting that the bark contains more extractives, and Hosseinaei et al. (2012) found that reducing their content limits the susceptibility of WPC to mold growth” 302-309
- It is not explained in this manuscript that why particles size is affecting fouling rate differently for PLA and HDPE?
It’s due to an overlap of two properties: filler size and porosity - PLA is generally more porous, which enhance penetration by mold, also in PLA the variation in filler size played a lesser role. PLA itself is better fouled than HDPE.
Reviewer 2 Report
It is an interesting manuscript, it discussed the factors of fungi growth in detail.
Author Response
We thank for professional and favorable review.
Reviewer 3 Report
The manuscript 1461285 describes resistance to mold about a woody plastic that worth as new bio-composites. My comments for the manuscript are as follows.
- Line 82: Please clarify the type of tree (pine, cedar, ----) and why select the type. Because, I think the effect on mold will differ depend on the type.
- Line103: “300*300*5m3” must be collected to “300*300*2.5m3”
- Table 4, 5, 6: Concerning to numerical notation, please use the period “.” as the decimal point not comma. Because they are percentage. Table 3 is right notation.
- Line 189: Authors write “porosity”, but readers cannot image then, so please show some SEM photographs of both PLA and HDPE composites.
- Figure 2, 3, 4, 5, 6: Each photos are distinguished by A and B, but please use (a) and (b) according to the caption of each figures.
- Line243-246: Authors discuss about the size effect on the accessible to fungi qualitatively, but authors should better more quantitatively discussion like a “area ratio per unit volume” , mean values of particle size and distribution.
- Line 247-254: Authors discuss trees as a nutrient for fungi, which is also an important aspect. On the other hand, I think the authors need discuss as antibacterial properties. For example, Cypresses are said to have very strong antibacterial properties against various microorganisms. Park wood also has antibacterial properties.
That's my comments.  14/December/2021
Author Response
We thank for professional comments on the manuscript. The manuscript has been revised according to the comments and suggestions. Each comment and its corresponding response are outlined below, including a sample of manuscript text where the modifications were applied, as well as a reference to page and line numbers in the revised manuscript.
- Line 82: Please clarify the type of tree (pine, cedar, ----) and why select the type. Because, I think the effect on mold will differ depend on the type.
The research results presented in the manuscript are a part of the research included in a larger project which aim was to develop directions and methods for the management of waste materials from sawmills and mechanical wood processing plants. Therefore, the selection of the type of wood and the type of raw material (including particle size) was dependent on the plants’ production profile that participated in the project.
- Line103: “300*300*5m3” must be collected to “300*300*5m3”
Thank you for that comment, the change has been made.
- Table 4, 5, 6: Concerning to numerical notation, please use the period “.” as the decimal point not comma. Because they are percentage. Table 3 is right notation.
Thank you for this comment, the numerical notation has been modified according to the reviewer’s suggestion.
- Line 189: Authors write “porosity”, but readers cannot image then, so please show some SEM photographs of both PLA and HDPE composites.
Porosity refers to the structure of the composite (both thermoplastic and filler) and photos in this area give an ambiguous picture (it is difficult to visualize in the SEM photo). The SEM photo shows either the filler particle or the thermoplastic itself. Therefore, the size of pores was omitted, instead was presented the share percentage of pores in the structure of the composite.
- Figure 2, 3, 4, 5, 6: Each photos are distinguished by A and B, but please use (a) and (b) according to the caption of each figures.
Thank you for this comment, photos markings and captions are now consistent..
- Line243-246: Authors discuss about the size effect on the accessible to fungi qualitatively, but authors should better more quantitatively discussion like a “area ratio per unit volume” , mean values of particle size and distribution.
The statement that mold fouling of the tested composites may be influenced by the accessibility of the surface of lignocellulosic particles is only a presumption, which, in the opinion of the authors, may explain the observed phenomena. Analytical determination of the size of the available lignocellulosic surfaces in the produced composites is a complex task involving many variables and, in the opinion of the authors of the study, may be an issue for a separate publication.
- Line 247-254: Authors discuss trees as a nutrient for fungi, which is also an important aspect. On the other hand, I think the authors need discuss as antibacterial properties. For example, Cypresses are said to have very strong antibacterial properties against various microorganisms. Park wood also has antibacterial properties.
In the case of pine or its bark, it is difficult to talk about antibacterial properties. Undoubtedly, the analysis of this type of properties of the produced composites is an interesting issue, but the performance of this type of analysis must be justified by the properties of the raw materials used to produce the composites. Therefore, in the case of pine and bark, the authors did not consider this type of research important.
Round 2
Reviewer 1 Report
Now, the manuscript can be accepted in the present form.